# Synthesis, In Silico Prediction and In Vitro Evaluation of Antitumor Activities of Novel Pyrido[2,3-*d*]pyrimidine, Xanthine and Lumazine Derivatives

**DOI:** 10.3390/molecules25215205

**Published:** 2020-11-09

**Authors:** Samar El-Kalyoubi, Fatimah Agili

**Affiliations:** 1Department of Pharmaceutical Organic Chemistry, Faculty of Pharmacy (Girls), Al-Azhar University, Nasr City, Cairo 11651, Egypt; 2Chemistry Department, Faculty of Science (Female Section), Jazan University, Jazan 82621, Saudi Arabia; fatmah2000@gmail.com

**Keywords:** 6-amino-1-benzyluracil, 6-Amino-1-(2-chlorobenzyl)uracil, 5,6-diamino-1-benzyl uracil, ethyl 5-aryl-7-methylpyridopyrimidine-6-carboxylate, 8-aryl-3-(2-chlorobenzyl) xanthines and 6-aryl-1-(2-chlorobenzyl)lumazines, cancer, molecular docking

## Abstract

Ethyl 5-arylpyridopyrimidine-6-carboxylates **3a**–**d** were prepared as a one pot three component reaction via the condensation of different aromatic aldehydes and ethyl acetoacetate with 6-amino-1-benzyluracil **1a** under reflux condition in ethanol. Additionally, condensation of ethyl 2-(2-hydroxybenzylidene) acetoacetate with 6-amino-1-benzyluracil in DMF afforded 6-acetylpyridopyrimidine-7-one **3e**; a facile, operationally, simple and efficient one-pot synthesis of 8-arylxanthines **6a**–**f** is reported by refluxing 5,6-diaminouracil **4** with aromatic aldehydes in DMF. Moreover, 6-aryllumazines **7a**–**d** was obtained via the reaction of 5,6-diaminouracil with the appropriate aromatic aldehydes in triethyl orthoformate under reflux condition. The synthesized compounds were characterized by spectral (^1^H-NMR, ^13^C-NMR, IR and mass spectra) and elemental analyses. The newly synthesized compounds were screened for their anticancer activity against lung cancer A549 cell line. Furthermore, a molecular-docking study was employed to determine the possible mode of action of the synthesized compounds against a group of proteins highly implicated in cancer progression, especially lung cancer. Docking results showed that compounds **3b**, **6c**, **6d**, **6e**, **7c** and **7d** were the best potential docked compounds against most of the tested proteins, especially CDK2, Jak2, and DHFR proteins. These results are in agreement with cytotoxicity results, which shed a light on the promising activity of these novel six heterocyclic derivatives for further investigation as potential chemotherapeutics.

## 1. Introduction

Pulmonary cancer among the two sexes has been the leading cause of cancer lung death for many years and worldwide statistics on incidence and mortality vary widely [1,2,3]. The world’s high mortality rate from malignant tumors is attributed to the uncontrolled growth of cells in lung tissues and to the high metastatic ability of lung cancer [4,5]. Tobacco consumption is a significant lung cancer risk factor. Genetic susceptibility, diet, alcohol consumption, industrial exposures and air pollution are other factors [6]. Pulmonary cancer is primarily caused by the abnormal growth of either small or non-small lung cancer cells [7,8,9,10]. In 2015, 80–85% of cases of lung cancer worldwide were attributed to non-small cell lung cancer (NSCLC) [11]. The main treatment for early lung cancer is surgical resection. Radiation is one of the most common methods of treating tumors. In case of advanced diseases (stage IIIB and stage IV), radiotherapy, targeted therapy and immunotherapy are applied [1]. Chemotherapy is one of the most common treatment methods for tumors [12,13]. However, current chemotherapeutic therapies appear to have several limitations, including a lack of tumor cell selectivity [14,15], significant toxic side effects on healthy tissue [16] and drug resistance [17] leading to unsatisfactory therapeutic effects.

The first form of antimetabolites introduced in the clinic 60 years ago was antifolate drugs [18,19]. Methotrexate (MTX) (**B**) is a folic acid (**A**) antagonist required for DNA synthesis, and has a therapeutic effect on many cancer cell types that over-express folate receptors on the surfaces of many of these cells (Figure 1) [20]. MTX inhibits the cellular folate metabolism by dihydrofolate reductase (DHFR) in a competitive way. By inhibiting the enzyme target DHFR [21], MTX suppresses purine and pyrimidine synthesis. It acts on S-phase and prevents DNA synthesis [22,23]. The chemotherapeutic MTX is widely used in human malignancies, including acute lymphoblastic leukemia, malignant lymphoma, lung cancer, breast cancer, osteosarcoma and head and neck cancer [24,25,26,27]. MTX has been also widely used for the treatment of rheumatoid arthritis (RA) through the release of adenosine-mediated suppression inflammation [28]. Clinical data supports the adenosine-mediated anti-inflammatory effect of MTX [29].

Chronic neurotoxicity can be caused by MTX [30]. It can also cause alveolitis and lung fibroblasting because of its pulmonary toxicity side effect [31,32]. Hoping to overcome the above mentioned chemotherapeutic limitations, a new series of pyridopyrimidines, xanthines and lumazines have been synthesized (Figure 1). All the newly synthesized compounds herein underwent a molecular-docking study and were screened for their anticancer activity against the A549 cell line.

## 2. Results

### 2.1. Chemistry

In this research, our strategy was directed towards developing new fused uracil derivatives of potential anticancer activity [33,34]. The desired starting materials, including 6-amino-1-benzyluracil **1a** [35] and 5,6-diamino-1-(2-chlorobenzyl)uracil **4** [36] were synthesized according to the literature by conventional methods.

Ethyl 5-aryl-2-thiopyridopyrimidine-6-carboxylates were prepared by heating under reflux of 6-amino-1-methyl-2-thiouracil with ethyl 3-aryl-2-cyanoacrylate in absolute ethanol in the presence of triethylamine (TEA) [37]. Ethyl 5-aryl-2-pyridopyrimidine-6-carboxylates **3a**–**d** were prepared in higher yields (60–84%) by refluxing compound **1a** with different aromatic aldehydes, ethyl acetoacetate in abs. ethanol containing TEA as a basic medium for 12 h as illustrated in (Scheme 1). The derivatives were crystallized from DMF/EtOH. Analogously, the treatment of compound **1a** with ethyl 2-(2-hydroxybenzylidene)acetoacetate in DMF in the presence of TEA for 4 h gave 6-acetyl-5-arylpyridopyrimidine 7-one **3e** (Scheme 1). The most interesting observation from comparing the ^1^H-NMR spectra of compounds **3a**–**d** and compound **3e** was the disappearance of both signals of CH-5 at δ 5.81 ppm and NH_2_ (6) at δ 5.97 ppm of the start **1a** and the appearance of ethyl protons at position 6 in pyridopyrimidines **3a**–**d** as triplet at δ 1.09–0.96 ppm for CH_3_ and quartet at δ 4.18–3.94 for CH_2_. A singlet signal occurred at δ 2.90 ppm for CH_3_ protons of acetyl group at position 6 in pyridopyrimidine **3e** and a singlet signal at δ 7.88 ppm for NH (8). On the other hand, the IR spectra of these pyridopyrimidines, which displayed the stretching band of the two C=O groups (Amide I), was red-shifted within the range υ 1745–1723, 1684–1697 cm^−1^. We noted the strong asymmetric and symmetric NO_2_ stretching bands of the nitro group at υ 1551 and 1374 cm^−1^, respectively, in compound **3d**. ^13^C-NMR spectra proves without doubt the formation of compounds **3a**–**d** through the appearance of the upfield signals of ethyl group of ester at δ 14.02–13.43 ppm for CH_3_ and at δ 61.81–59.55 ppm for CH_2_, in addition to the presence of a signal at δ 18.13–17.52 ppm for CH_3_(7). While in comparison with the ^13^C-NMR spectra of compound **3e**, it was observed that the signal of CH_3_ of acetyl group was δ 17.53 ppm.

The mechanism outlined in Scheme 2 seems to be the most plausible pathway through the Michael addition reaction via the formation of non-isolated Michael adduct intermediate that underwent intramolecular cyclization in protic solvent affording the corresponding **3a**–**d**. On the other hand, Scheme 3 illustrates the mechanistic pathway for the formations of compound **3e** via the formation of Michael adduct intermediate followed by cyclocondensation in aprotic solvent.

8-Aryl xanthines have been reported [36,38]. Herein, condensation of 5,6-diaminouracil **4** with different aromatic aldehydes in DMF under reflux for two to three hours afforded 8-aryl xanthine derivatives **6a**–**f** (Scheme 4). The reaction proceeded through the nucleophilic attack of the amino group at position 5 of uracil **1b** to the electrophilic center of the carbonyl group of the aromatic aldehydes followed by dehydration to form the intermediate Schiff base which undergoes intramolecular cyclization to afford **6a**–**f** in good to excellent yield. The intrinsic significance of the IR spectra is that the presence of the stretching band of the 2 C=O groups (Amide I) within the range υ 1684, 1643 cm^−1^. ^1^H-NMR spectra prove the suggested xanthines **6a**–**f** by the presence of both characteristic singlet signals for NH(9) at δ 14.46–13.23 ppm and NH(1) at δ 1139–10.94 ppm. The phenyl group protons appeared at δ 8.60–6.81 ppm. The clearance of the spectra from the signals at δ 6.78 ppm and at δ 5.79 ppm, characteristic for 2 NH_2_ protons of uracil **4** at position 5 and 6, respectively, supported the previous observation. Also, the ^13^C-NMR spectra and Mass spectra supported the previous expectation. With these results in hand, xanthines **6a**–**f** became confirmed without a doubt.

Moreover, heating of 5,6-diaminouracil **4** with various aromatic aldehydes in triethyl orthoformate under reflux for 7–8 h afforded 6-aryllumazines **7a**–**d** (Scheme 4). Structures **7a**–**d** were inferred from their spectral data, and elemental analyses. ^1^H-NMR data for compound **7a**–**d** showed a singlet signal at δ 12.01–11.29 ppm for NH-3 and a singlet signal at δ 8.71–8.70 ppm for CH-7 in lumazine ring. The IR spectrum revealed bands at the range of υ 3141–3127 cm^−1^ characteristic for NH. Compound **7b** showed a characteristic band for OH at υ 3480 cm^−1^. The mechanistic pathway is illustrated via the reaction of the intermediate Schiff base **5** with triethyl orthoformate followed by intramolecular cycloaddition accompanied by elimination of EtOH (Scheme 5). The plausible mechanism was proved by isolating the intermediate Schiff base **5** with its interesting spectral data. Otherwise, compound **5** was refluxed with DMF afforded xanthine **6c**, while refluxing with triethyl orthoformate afforded the lumazine **7b**.

### 2.2. Biological Activity

#### Anticancer Evaluation

During initial cytotoxic screening for the tested compounds, we tested the activity of the compounds against two cell lines, which were lung (A549 cell line) and breast cancer (MCF7). The cytotoxic activity against lung cancer was potentially more promising than breast cancer. Therefore, we have selected the lung cancer cell line.

All the newly synthesized compounds were screened for their cytotoxic effects on the human cell lung adenocarcinoma A549 cell line. The obtained IC_50_ values of the synthesized compounds were compared to well-known reference anticancer drug methotrexate under the same conditions using a colorimetric viability assay. IC_50_ values were determined from plots of a dose response curve of the concentration of test compounds required to kill 50% of cell population. The in vitro growth inhibition results showed that all the tested compounds have an inhibitory effect on the tumor cell line A549 in a concentration dependent manner. All the results are represented in Table 1 and Figure 2.

It was noticed that compounds **3b** and **7d** exhibited the highest inhibitory activity against the Lung carcinoma cell line (A549) in comparison with the reference drug methotrexate with IC_50_ values of 10.3 and 12.2 μM, respectively. Also, compounds **6c**–**6e** and **7c** showed potent antitumor activity against the Lung carcinoma cell line (A549) with IC_50_ values of 23.1, 26.3, 27 and 24.9 μM, respectively. The other compounds have less antitumor activity (Table 1 and Figure 2).

### 2.3. Computer-Aided Docking

To take one step further to determine the mode of action of the tested compounds, a molecular docking study was employed to determine the binding modes against series of proteins such as CDK2, BCL2, Jak2, P53 binding site in MDM2, and DHFR that are implicated significantly in cancer disease (Appendix A). These targets were selected based on their potential roles in apoptosis regulation and limiting lung cancer progression, therefore, targeting these macromolecules provides potential benefits in lung cancer therapy. The co-crystal ligands were re-docked to assure the validity of the docking parameters and methods used to represent the position and orientation of the ligand detected in the crystal structure. The difference of RMSD value between co-crystal ligands to the original co-crystal ligand was <2 Å which approved the accuracy of the docking protocols and parameters [39].

The docking results depicted that all tested compounds showed poor binding affinity with BCL-2 protein, as the binding free energy of all tested compounds was higher than the binding free energy of co-crystalized ligands (ΔG_b_ = −10.6 kcal/mol). In the same line, all tested compounds had low binding affinity with MDM2-P53 protein, except compounds **6d** and **7c**, as they had lower binding free energy (−8.3 kcal/mol, and −8.4 kcal/mol, respectively) compared to [(3*R*,5*R*,6*S*)-5-(3-chlorophenyl)-6-(4-chlorophenyl)-1-(cyclopropylmethyl)-2-oxopiperidin-3-yl] acetic acid which is abbreviated as **13Q** the co-crystalized ligand (ΔG_b_ = −8.2 kcal/mol) (Table 2) (Appendix A). Based on these results, the mode of action of the tested compound might not be related to both BCL-2 and MDM2-P53 proteins. All tested compounds showed high binding affinity against CDK2 proteins, as all tested compounds showed low binding free energies compared to both reference drug (ΔG_b_ = −8.3 kcal/mol) and methotrexate (ΔG_b_ = −7.6 kcal/mol), which implies a potential inhibitory effect of the tested compounds against the cyclin dependent kinase enzyme.

Based on the molecular docking study, we tested the proposed activity of the tested compounds against DHFR protein, which is the main target for MTX. Compounds **3b**, **6c**, **6d**, **6e**, **7c** and **7d** were the best docked compounds against CDK2, Jak2, and DHFR proteins, which are displayed by low binding free energy (Table 2), hydrogen bond formation (Table 3, Table 4 and Table 5), and hydrophobic interaction (Figure 3 and Figure 4, Appendix A) compared to reference ligand and methotrexate. These results are in agreement with the cytotoxicity assay as compounds **3b**, **6c**, **6d**, **6e**, **7c** and **7d** showed lower IC_50s_ (10.3 ± 0.2, 27.0 ± 1.1, 23.1 ± 0.6, 26.3 ± 1.3, 24.9 ± 1.2, 12.2 ± 0.3 µM, respectively) compared to methotrexate as a reference drug (36.3 ± 3.9 µM).

## 3. Experimental Part

### 3.1. Chemistry

All melting points were determined with an Electrothermal Mel.-Temp. II apparatus and were uncorrected. Element analyses were performed at the Regional Center for Mycology and Biotechnology at Al-Azhar University. IR spectra were obtained in the solid state in the form of KBr discs using a Perkin-Elmer Model 1430 spectrometer (Perkin-Elmer, Akron, OH, USA) and carried out in Taif University, Taif, Saudi Arabia. The proton nuclear magnetic resonance (^1^H-NMR) spectra were recorded on Varian Gemini 400 MHz Spectrometer using DMSO-*d*_6_ as a solvent (Chemical shift in δ, ppm), Faculty of Science, Chemistry Department, Zagazig University, Zagazig, Egypt. Mass spectra were recorded on a DI-50 unit of Shimadzu GC/MS-QP 5050A (Kyoto, Japan) at the Regional Center for Mycology and Biotechnology at Al-Azhar University, Cairo, Egypt. All reactions were monitored by TLC using precoated plastic sheets silica gel (Merck 60 F254) and spots were visualized by irradiation with UV light (254 nm). The used solvent system was chloroform: Methanol (9:1) & ethyl acetate: Toluene (1:1). 6-amino-1-benzyluracil **1a** was prepared as the reported method [35], 6-diamino-1-(2-chlorobenzyl)uracil **1b** was prepared as the reported method [36] and 5,6-diamino-1-(2-chlorobenzyl)uracil **4** was prepared as the reported method [36].

#### 3.1.1. Ethyl 5-Aryl-1-benzyl-7-methyl-2,4-dioxo-1,2,3,4-tetrahydropyrido[2,3-*d*]pyrimidine-6-carboxylate **3a**–**d**

General method: A mixture of 6-amino-1-benzyluracil (0.3 g, 1.38 mmol), ethyl acetoacetate (1.38 mmol) and appropriate benzaldehyde derivatives (1.38 mmol) in ethanol (5 mL) in the presence of TEA (1.0 mL). The reaction mixture was heated under reflux for 12 h. The formed precipitate was filtered off, washed with ethanol and recrystallized from DMF/ethanol (2:1) and dried to give the desired compounds **3a**–**d** (Scheme 1).

*Ethyl 1-benzyl-5-phenyl-7-methyl-2,4-dioxo-1,2,3,4-tetrahydropyrido[2,3-d]pyrimidine-6-carboxylate* (**3a**): Yield: 66%; m.p.: 201–203 °C; IR (KBr) ν_max_ (cm^−1^): 3186 (NH), 3055 (CH arom), 2920, 2847 (CH aliph), 1730, 1682 (2C=O), 1595 (NH bending), 694,748 (monosubstituted phenyl); ^1^H-NMR (DMSO-*d*_6_): δ 11.81 (s, 1H, NH), 7.47–7.43 (m, 5H, arom), 7.30–7.22 (m, 5H, arom), 5.39 (s, 2H, CH_2_), 4.15–4.09 (q, 2H, CH_2_), 2.72 (s, 3H, CH_3_(7)), 0.99–0.96 (t, 3H, CH_3_); MS: *m*/*z* (%) = M^+^, 415 (100), 400 (11), 370 (9), 358 (8), 344 (10), 341 (7), 310 (14), 297 (3), 91 (14), 65 (3); Anal. calcd. for C_24_H_21_N_3_O_4_ (415.44): C, 69.39; H, 5.10; N, 10.11. Found: C, 69.13; H, 5.28; N, 10.34.

*Ethyl 1-benzyl-5-(2-chlorophenyl)-7-methyl-2,4-dioxo-1,2,3,4-tetrahydropyrido[2,3-d]pyrimidine-6-carboxylate* (**3b**): Yield: 60%; m.p.: 258–260 °C; IR (KBr) ν_max_ (cm^−1^): 3165 (NH), 3034 (CH arom), 2920, 2840 (CH aliph), 1745, 1697 (2C=O), 1603 (NH bending), 754 (*o*.substituted phenyl); ^1^H-NMR (DMSO-*d*_6_): δ 10.92 (s, 1H, NH), 7.25–7.22 (m, 4H, arom), 7.11–7.03 (m, 3H, arom), 6.87–6.85 (d, 2H, *J* = 7.2 Hz, arom.), 5.18 (s, 2H, CH_2_), 4.04–3.94 (q, 2H, *J* = 6.8, CH_2_), 2.64 (s, 3H, CH_3_(7)), 1.09–1.06 (t, 3H, *J* = 6.8, CH_3_); ^13^C-NMR (DMSO-*d*_6_): δ = 14.02, 17.52, 43.66, 59.55, 109.24, 125.82, 125.92, 126.97, 127.39, 128.17, 128.45, 129.57, 129.65, 131.32, 135.34, 137.72, 146.17, 149.76, 151.53, 151.56, 160.17, 165.20; MS: *m*/*z* (%) = M + 2, 452 (18), M^+^, 450 (76), 424 (34), 358 (2), 341 (7), 339 (7), 258 (3), 152 (2), 92 (5), 91 (100), 65 (11); Anal. calcd. for C_24_H_20_ClN_3_O_4_ (449.88): C, 64.07; H, 4.48; N, 9.34. Found: C, 63.89; H, 4.62; N, 9.51.

*Ethyl 1-benzyl-5-(4-bromophenyl)-7-methyl-2,4-dioxo-1,2,3,4-tetrahydropyrido[2,3-d]pyrimidine-6-carboxylate* (**3c**): Yield: 73%; m.p.: 217–219 °C; IR (KBr) ν_max_ (cm^−1^): 3117 (NH), 3031 (CH arom), 2948, 2845 (CH aliph), 1725, 1681 (2C=O), 1558 (NH bending), 838 (*p*.substituted phenyl); ^1^H-NMR (DMSO-*d*_6_): δ 11.82 (s, 1H, NH), 7.70–7.68 (d, 2H, *J* = 8.4 Hz, arom), 7.38–7.36 (d, 2H, *J* = 8.4 Hz, arom), 7.30–7.29 (d, 2H, *J* = 4.4 Hz, arom.), 7.23–7.18 (m, 3H, arom), 5.3 (s, 2H, CH_2_), 4.19–4.13 (q, 2H, *J* = 6.8, CH_2_), 2.73 (s, 3H, CH_3_(7)), 1.05–1.02 (t, 3H, *J* = 6.8, CH*3*); ^13^C-NMR (DMSO-*d*_6_): δ = 13.43, 18.03, 44.62, 61.78, 108.26, 123.65, 125.39, 126.95, 127.14, 128.29, 130.01, 131.61, 137.26, 137.44, 150.30, 150.92, 151.87, 156.08, 161.53, 167.32; MS: *m*/*z* (%) = M + 2, 496 (12), M^+^, 494 (32), 490 (23), 387 (34), 341 (11), 292 (59), 265 (62), 129 (58), 72 (100), 69 (92); Anal. calcd. for C_24_H_20_ClN_3_O_4_ (494.33): C, 58.31; H, 4.08; N, 8.50. Found: C, 58.57; H, 4.23; N, 8.69.

*Ethyl 1-benzyl-7-methyl-5-(3-nitrophenyl)-2,4-dioxo-1,2,3,4-tetrahydropyrido[2,3-d]pyrimidine-6-carboxylate* (**3d**): Yield: 84%; m.p.: 190–192 °C; IR (KBr) ν_max_ (cm^−1^): 3215 (NH), 3050 (CH arom), 2965, 2869 (CH aliph), 1723, 1684 (2C=O), 1614 (C=C), 1551, 1374 (NO_2_), 755 (*m*.substituted phenyl); ^1^H-NMR (DMSO-*d*_6_): δ 11.90 (bs, 1H, NH), 8.25 (s, 1H, arom),7.94–7.39 (m, 3H, arom), 7.32–7.16 (m, 5H, arom), 5.24 (s, 2H, CH_2_), 4.18–4.14 (q, 2H, *J* = 7.2 Hz, CH_2_), 2.88 (s, 3H, CH_3_(7)), 1.04–1.01 (t, 3H, *J* = 7.2 Hz, CH_3_); ^13^C-NMR (DMSO-*d*_6_): δ = 13.41, 18.13, 44.78, 61.81, 107.90, 123.18, 125.23, 126.94, 127.03, 127.22, 127.59, 128.29, 131.61, 136.43, 137.26, 137.44, 150.50, 150.98, 151.84, 156.12, 161.60, 167.45; MS: *m*/*z* (%) = M^+^, 460 (13), 430 (70), 391 (51), 360 (27), 345 (28), 343 (18), 234 (31), 209 (64), 182 (29), 173 (100), 172 (29), 69 (92); Anal. calcd. for C_24_H_20_N_4_O_6_ (460.43): C, 62.60; H, 4.38; N, 12.17. Found: C, 62.86; H, 4.54; N, 12.41.

*6-Acetyl-1-benzyl-5-(2-hydroxyphenyl)pyrido[2,3-d]pyrimidine-2,4,7(1H,3H,8H)-trione* (**3e**): A mixture of 6-amino-1-benzyluracil (0.5 g, 2.3 mmol), ethyl 2-(2-hydroxybenzylidene) acetoacetate (2.3 mmol) was mixed in DMF (1.5 mL) in the presence of TEA (1.0 mL). The reaction mixture was heated under reflux for 7 h. The mixture was evaporated in vacuo and the residue was treated with methanol (10 mL). The formed precipitate was filtered off and recrystallized from DMF/ethanol (2:1) to afford **3e**. Yield: 68%; m.p.: 260–262 °C; IR (KBr) ν_max_ (cm^−1^): 3448 (OH), 3186 (NH), 3047 (CH arom), 2924, 2846 (CH aliph), 1681 (C=O), 1589 (NH bending), 763 (*o*.substituted phenyl); ^1^H-NMR (DMSO-*d*_6_): δ13.98 (s, 1H, OH), 11.98 (s, 1H, NH), 8.19–8.17 (d, 1H, arom), 7.69 (s, 1H, NH 8), 7.67–7.61 (m, 3H, arom), 7.43–7.21 (m, 5H, arom), 5.45 (s, 2H, CH2), 2.90 (s, 3H, CH_3_(6)); ^13^C-NMR (DMSO-*d*_6_): δ = 17.52, 43.66, 103.73, 112.40, 116.16, 122.65, 124.73, 127.10, 127.75, 128.33, 132.03, 133.62, 137.22, 146.90, 150.29, 151.89, 153.86, 158.19, 160.37, 167.14; MS: *m*/*z* (%) = M^+^, 403 (84), 361 (29), 315 (24), 268 (19), 250 (100), 228 (23), 186 (48), 125 (87), 76 (53), 56 (30); Anal. calcd. for C_22_H_17_N_3_O_5_ (403.38): C, 65.50; H, 4.25; N, 10.42. Found: C, 65.89; H, 4.32; N, 10.51.

*6-Amino-1-(2-chlorobenzyl)-5-[(E)-(4-chlorobenzylidene)amino]pyrimidine-2,4(1H,3H)-dione* (**5**): A mixture of 5,6-diamino-1-(2-chlorobenzyl)uracil (0.8 g, 2.0 mmol), 4-chlorobenzaldehyde (0.2 mmol) in DMF (3.0 mL) was heated under reflux for 45 min. After cooling, the formed precipitate was collected by filtration, washed with ethanol and dried in an oven to give (1.01 g). Yield: 87%; m.p.: 283–285 °C; IR (KBr) ν_max_ (cm^−1^): 3339, 3274 (NH_2_), 3184 (NH), 3078 (CH arom), 1691, 1643 (2 C=O), 822 (*p*.substituted phenyl), 748 (C-Cl); ^1^H-NMR (DMSO-*d*_6_): δ 10.92 (s, 1H, NH 3), 9.72 (s, 1H, CH), 7.52–7.50 (d, 2H, *J* = 8.4 Hz, arom.), 7.44–7.34 (m, 4H, arom), 7.27–7.25 (d, 2H, *J* = 8.4 Hz, arom), 6.99 (s, 2H, NH_2_ 6), 5.24 (s, 2H, CH_2_); MS: *m*/*z* (%) = M + 2, 391 (4), M^+^, 389 (14), 375 (5), 263 (10), 246 (92), 216 (30), 190 (13), 125 (100), 89 (59), 63 (18); Anal. calcd. for C_18_H_14_Cl_2_N_4_O_2_ (389.23): C, 55.54; H, 3.63; N, 14.39. Found: C, 55.62; H, 3.65; N, 14.53.

#### 3.1.2. 8-Aryl-3-(2-chlorobenzyl)-xanthine

##### 8-Aryl-3-(2-chlorobenzyl)-3,9-dihydro-1H-purine-2,6-dione **6a**–**f**

General method: Method A: To 5,6-diamino-1-(2-chlorobenzyl)uracil **4** (0.5 mmol), was added different aromatic aldehydes (0.5 mmol) in DMF (3.0 mL). The reaction mixture was heated under reflux for 2–3 h. After cooling, the formed precipitate was filtered off, washed with methanol and recrystallized from DMF/H_2_O 3:1) to give the desired compounds **6a**–**f** in good yields. Method B: Compound **5** (0.2 g, 0.5 mmol) in DMF (3 mL) was heated under reflux for 90 min. After cooling, the formed precipitate was collected by filtration, washed with methanol and dried in an oven to afford **6c** in a good yield.

#### 3.1.3. 3-(2-Chlorobenzyl)-8-phenylxanthine

*3-(2-chlorobenzyl)-8-phenyl-3,9-dihydro-1H-purine-2,6-dione* (**6a**): Yield 83%; m.p.: >300 °C; IR (KBr) ν_max_ (cm^−1^): 3334, 3224 (2NH), 3078 (CH arom), 2970, 2831 (CH aliph,), 1667 (C=O), 748 (C-Cl); ^1^H-NMR (DMSO-*d*_6_): δ 13.23 (s, 1H, NH 9), 10.94 (s, 1H, NH 1), 7.95–7.61 (m, 2H, arom.), 7.51–7.48 (m, 3H, arom), 7.35–7.32 (m, 3H, arom), 6.88–6.84 (m, 1H, arom), 5.07 (s, 2H, CH_2_); ^13^C-NMR (DMSO-*d*_6_): δ = 43.0, 107.80, 125.39, 127.49, 127.50, 128.60, 128.69, 129.38, 131.46, 131.61, 133.31, 133.69, 149.78, 149.99, 154.62, 162.29; MS: *m*/*z* (%) = M + 2, 354 (4), M^+^, 352 (16), 331 (26), 312 (29), 286 (30), 239 (29), 110 (28), 98 (31), 84 (35), 83 (100); Anal. calcd. for C_18_H_13_ClN_4_O_2_ (352.78): C, 61.28; H, 3.71; N, 15.88. Found: C, 61.47; H, 3.87; N, 15.74.

##### 3-[(2-Chlorophenyl)methyl]-8-(4-hydroxyphenyl)-3,9-dihydro-1*H*-purine-2,6-dione

*3-(2-Chlorobenzyl)-8-(4-hydroxyphenyl)xanthine* (**6b**):Yield: 57%; m.p.: >300 °C; IR (KBr) ν_max_ (cm^−1^): 3450 (OH), 3330, 3142 (2NH), 3027 (CH arom), 2922, 2819 (CH aliph.), 1667 (C=O), 1599 (NH bending), 841 (*p*. substituted phenyl), 766. (C–Cl); ^1^H-NMR (DMSO-*d*_6_): δ 13.55 (s, 1H, NH 9), 11.21 (s, 1H, NH 1), 9.96 (s, 1H, OH), 7.89–7.87 (d, 2H, *J* = 6.8 Hz, arom.), 7.31–7.25 (m, 3H, arom), 7.04–7.02 (m, 1H, arom), 6.83–6.81 (d, 2H, *J* = 6.8, arom), 5.23 (s, 2H, CH_2_); ^13^C-NMR (DMSO-*d*_6_): δ = 43.0, 107.6, 115.7, 119.57, 127.06, 127.48, 128.21, 128.78, 129.34, 131.51, 134.15, 149.87, 150.95, 154.62, 159.45, 163,29; MS: *m*/*z* (%) = M + 2, 370 (2), M^+^, 368 (4), 333 (22), 259 (24), 216 (12), 125 (100), 89 (26), 77 (3); Anal. calcd. for C_18_H_13_ClN_4_O_3_ (368.77): C, 58.62; H, 3.55; N, 15.19. Found: C, 58.74; H, 3.58; N, 15.34.

##### 3-[(2-Chlorophenyl)methyl]-8-(4-chlorophenyl)-3,9-dihydro-1*H*-purine-2,6-dione

*3-(2-Chlorobenzyl)-8-(4-chlorophenyl)xanthine* (**6c**): Method A: Yield: 67% Method B: Yield: 85%; m.p.: >300 °C; IR (KBr) ν_max_ (cm^−1^): 3311, 3142 (2NH), 3022 (CH arom), 2987, 2838 (CH aliph), 1682 (C=O), 1583 (NH bending), 830 (*p*.substituted phenyl), 740 (C-Cl); ^1^H-NMR (DMSO-*d*_6_): δ 13.99 (s, 1H, NH 9), 11.33 (s, 1H, NH 1), 8.06–8.04 (d, 2H, *J* = 6.8 Hz, arom), 7.55–7.53 (d, 2H, *J* = 6.8 Hz, arom), 7.30–7.07 (m, 3H, arom.), 7.06–7.05 (d, 1H, *J* = 1.6 Hz, arom), 5.24 (s, 2H, CH_2_); ^13^C-NMR (DMSO-*d*_6_): δ = 43.06, 108.6, 127.11, 127.46, 128.02, 128.79, 129.08, 129.33, 131.49, 134.00, 134.83, 149.65, 150.89, 150.98, 154.69, 163.20; MS: *m*/*z* (%) = M + 4, 391 (0.25), M+2, 389 (0.75), M^+^, 387 (2), 353 (13), 351 (45), 175 (4), 138 (11), 127 (26), 125 (100), 89 (24); Anal. calcd. for C_18_H_12_Cl_2_N_4_O_2_ (387.21): C, 55.83; H, 3.12; N, 14.47. Found: C, 56.01; H, 3.19; N, 14.62.

##### 3-[(2-Chlorophenyl)methyl]-8-(2-hydroxyphenyl)-3,9-dihydro-1*H*-purine-2,6-dione

*3-(2-Chlorobenzyl)-8-(2-hydroxyphenyl)xanthine* (**6d**): Yield: 58%; m.p.: >300; IR (KBr) ν_max_ (cm^−1^): 3480 (OH), 3320, 3168 (2NH), 3026 (CH arom), 2910, 2831 (CH aliph,), 1668, 1643 (2C=O), 1583 (NH bending), 747 (*o*.substituted phenyl); ^1^H-NMR (DMSO-*d*_6_): δ 13.85 (s, 1H, NH 9), 11.41 (s, 1H, NH 1), 9.55 (s, 1H, OH), 8.01–7.99 (d, 1H, arom.), 7.52–7.50 (m, 2H, arom), 7.31–7.25 (m, 3H, arom), 7.17–6.90 (m, 2H, arom), 5.26 (s, 2H, CH_2_); ^13^C-NMR (DMSO-*d*_6_): δ = 42.30 (CH2), 108.10, 123.00, 125.20, 127.25, 127.55, 127.75, 128.16, 129.07, 129.38, 129.49, 131.30, 134.28, 149.79, 150.96, 154.21, 161.67, 162.33; MS: *m*/*z* (%) = M + 2, 370 (2), M^+^, 368 (0.98), 290 (4), 259 (34), 172 (21), 152 (5), 125 (100), 107 (16), 89 (28), 77 (11); Anal. calcd. for C_18_H_13_ClN_4_O_3_ (368.77): C, 58.62; H, 3.55; N, 15.19. Found: C, 58.72; H, 3.59; N, 15.31.

##### 3-[(2-Chlorophenyl)methyl]-8-(4-nitrophenyl)-3,9-dihydro-1*H*-purine-2,6-dione

*3-(2-Chlorobenzyl)-8-(4-nitrophenyl)xanthine* (**6e**): Yield: 71%; m.p.: >300; IR (KBr) ν_max_ (cm^−1^): 3329, 3155 (2NH), 3042 (CH arom), 2926, 2822 (CH aliph), 1685, 1664 (2C=O), 1573 (NH bending), 1518, 1338 (NO_2_), 854 (*p*.substituted phenyl), 745 (C-Cl); ^1^H-NMR (DMSO-*d*_6_): δ 14.36 (s, 1H, NH 9), 11.39 (s, 1H, NH), 8.32–8.30 (d, 2H, *J* = 8.4 Hz, arom), 7.52–7.50 (d, 2H, *J* = 8.4 Hz, arom), 7.33–7.25 (m, 3H, arom), 7.09–7.07 (d, 3H, *J* = 7.2, arom.), 5.25 (s, 2H, CH_2_); ^13^C-NMR (DMSO-*d*_6_): δ = 43.45, 107.90, 124.25, 127.12, 127.46, 128.06, 128.30, 128.90, 129.34, 131.50, 134.82, 149.50, 150.90, 152.0, 154.23, 162.21; MS: *m*/*z* (%) = M + 2, 399 (0.46), M^+^, 397 (0.7), 324 (29), 283 (6), 166 (30), 150 (10), 127 (100), 91 (18), 67 (17); Anal. calcd. for C_18_H_12_ClN_5_O_4_ (397.21): C, 54.35; H, 3.04; N, 17.61. Found: C, 54.52; H, 3.12; N, 17.89.

##### 8-(4-Bromophenyl)-3-[(2-chlorophenyl)methyl]-3,9-dihydro-1*H*-purine-2,6-dione

*8-(4-Bromophenyl)-3-(2-Chlorobenzyl)xanthine* (**6f**): Yield: 62%; m.p.: > 300 °C; IR (KBr) ν_max_ (cm^−1^): 3280, 3170 (2NH), 3091 (CH arom), 2998, 2802 (CH aliph), 1678, (C=O), 1583 (NH bending), 824 (*p*.substituted), 741 (C-Cl), 660 (C-Br); ^1^H-NMR (DMSO-*d*_6_): δ 14.03 (s, 1H, NH 9), 11.35 (s, 1H, NH), 7.99–7.97 (d, 2H, *J* = 8.4 Hz, arom), 7.69–7.67 (d, 2H, *J* = 8.4 Hz, arom), 7.32–7.23 (m, 3H, arom.), 7.07–7.05 (d, 1H, *J* = 7.2 Hz, arom.), 5.23 (s, 2H, CH_2_); ^13^C-NMR (DMSO-*d*_6_): δ = 42.90, 108.60, 127.08, 127.44, 128.17, 128.77, 130.69, 131.48, 131.94, 134.03, 136.24, 149.70, 150.90, 154.72, 157.32, 162.30; MS: *m*/*z* (%) = M + 4, 435 (0.87), M + 2, 433 (0.79), M^+^, 431 (5), 394 (5), 308 (14), 216 (13), 169 (19), 73 (100), 43 (12); Anal. calcd. for C_18_H_12_BrClN_4_O_2_ (431.67): C, 50.08; H, 2.80; N, 12.98. Found: C, 50.19; H, 2.77; N, 13.14.

##### 6-Aryl-1-[(2-chlorophenyl)methyl]pteridine-2,4(1*H*,3*H*)-dione

*6-Aryl-1-(2-chlorobenzyl)lumazine***7a**–**d**: General method: Method A: A mixture of 5,6-diaminouracil **4** (0.5 mmol), appropriate aromatic aldehydes (0.5 mmol) and triethyl orthoformate (4.5 mL) was heated under reflux for 7–8 h. After cooling, the formed precipitate was collected by filtration, recrystallized from DMF/EtOH (2:1). Method B: Compound **5** (0.2 g, 0.5 mmol) in triethyl orthoformate (5 mL) was heated under reflux for 4 h. After cooling, the formed precipitate was collected by filtration, washed with methanol and dried in an oven to afford **7a**–**d**.

##### 6-(4-Chlorophenyl)-1-[(2-chlorophenyl)methyl]pteridine-2,4(1*H*,3*H*)-dione

*6-(4-Chlorophenyl)-1-(2-chlorobenzyl)lumazine* (**7a**): Method A: Yield: 74%, Method B: Yield: 82%; m.p.: 250–252 °C; IR (KBr) ν_max_ (cm^−1^): 3141 (NH), 3070 (CH arom), 2976, 2860 (CH aliph), 1682, 1668 (2C=O), 1590 (NH bending), 898 (*p*.substituted phenyl), 748 (C-Cl); ^1^H-NMR (DMSO-*d*_6_): δ 11.99 (s, 1H, NH), 8.71 (s, 1H, CH-7), 7.28–7.26 (d, 2H, *J* = 8.0 Hz, arom), 7.11–6.96 (m, 4H, arom), 6.84–6.82 (d, 2H, *J* = 8.0 Hz, arom), 5.10 (s, 2H, CH_2_); ^13^C-NMR (DMSO-*d*_6_): δ = 42.46, 107.28, 126.09, 127.45, 128.24, 128.58, 129.21, 129.35, 131.09, 131.48, 133.90, 137.61, 150.39, 151.12, 154.11, 155.89, 162.33; MS: *m*/*z* (%) = M + 4, 403 (0.70), M + 2, 401, (1), M^+^, 399 (2.45), 328 (3), 190 (25), 122 (100), 93(46), 65 (48); Anal. calcd. for C_19_H_12_Cl_2_N_4_O_2_ (399.23): C, 57.16; H, 3.03; N, 14.03. Found: C, 57.34; H, 3.11; N, 14.29.

##### 1-[(2-Chlorophenyl)methyl]-6-(4-hydroxyphenyl)pteridine-2,4(1*H*,3*H*)-dione

*1-(2-chlorobenzyl)-6-(4-hydroxyphenyl)lumazine* (**7b**): Yield: 57%; m.p.: 280–282 °C; IR (KBr) ν_max_ (cm^−1^): 3480 (OH), 3127 (NH), 3025 (CH arom), 2914, 2856 (CH aliph), 1697, 1654 (2C=O), 1609 (C=C), 827 (*p*.substituted phenyl), 749 (C-Cl); ^1^H-NMR (DMSO-*d*_6_): δ 12.01 (s, 1H, NH), 10.53 (s,1H, OH), 8.71 (s, 1H, CH-7), 7.34–7.27 (m, 3H, arom), 7.13–7.09 (t, 1H, arom), 6.99–6.95 (t, 1H, arom), 6.87–6.83 (m, 3H, arom.), 5.10 (s, 2H, CH_2_); ^13^C-NMR (DMSO-*d*_6_): δ = 42.97, 107.29, 125.35, 126.86, 127.45, 128.25, 128.94, 129.35, 131.49, 133.90, 137.62, 150.39, 151.12, 154.11, 155.89, 160.27, 162.34; MS: *m*/*z* (%) = M + 2, 382 (0.58), M^+^, 380 (0.18), 281 (4), 270 (3), 253 (4), 242 (16), 241 (100), 210 (8), 198 (16), 125 (98), 99 (14), 89 (31), 71 (19); Anal. calcd. for C_19_H_13_ClN_4_O_3_ (380.78): C, 59.93; H, 3.44; N, 14.71. Found: C, 60.12; H, 3.49; N, 14.87.

##### 1-(2-Chlorobenzyl)-6-(2-hydroxyphenyl)pteridine-2,4(1*H*,3*H*)-dione

*1-(2-Chlorobenzyl)-6-(2-hydroxyphenyl)lumazine* (**7c**): Yield: 61%; m.p.: 252–254 °C; IR (KBr) ν_max_ (cm^−1^): 3423 (OH), 3179 (NH), 3051 (CH arom), 2911, 2850 (CH aliph), 1685 (C=O), 1583 (NH bending), 761 (*o*.substituted phenyl); ^1^H-NMR (DMSO-*d*_6_): δ 11.29 (s, 1H, NH), 10.25 (s,1H, OH), 8.70 (s, 1H, CH-7), 7.66–7.64 (d, 1H, arom), 7.53–7.50 (t, 1H, arom), 7.31–7.22 (m, 3H, arom.), 7.18–7.00 (m, 1H, arom), 6.98–6.82 (m, 2H, arom.), 5.09 (s, 2H, CH2); MS: *m*/*z* (%) = M + 2, 382 (17), M^+^, 380 (19), 359 (37), 354 (60), 229 (100), 217 (50), 177 (24), 83 (21), 60 (42); Anal. calcd. for C_19_H_13_ClN_4_O_3_ (380.78): C, 59.93; H, 3.44; N, 14.71. Found: C, 60.19; H, 3.57; N, 14.95.

##### 1-(2-Chlorobenzyl)-6-(4-nitrophenyl)pteridine-2,4(1*H*,3*H*)-dione

*1-(2-Chlorobenzyl)-6-(4-nitrophenyl)lumazine* (**7d**): Yield: 95%; m.p.: 276–278 °C; IR (KBr) ν_max_ (cm^−1^): 3179 (NH), 3034 (CH arom), 2912, 2858 (CH aliph), 1693 (C=O), 1600 (NH bending), 1548, 1397 (NO_2_), 869 (*p*.substituted phenyl), 751 (C-Cl); ^1^H-NMR (DMSO-*d*_6_): δ 11.99 (s, 1H, NH), 8.70 (s, 1H, CH-7), 7.50–7.48 (d, 2H, arom), 7.31–7.22 (m, 4H, arom), 6.96–6.94 (d, 2H, arom), 5.09 (s, 2H, CH_2_); ^13^C-NMR (DMSO-*d*_6_): δ = 43.00, 107.28, 126.08, 126.89, 127.39, 128.23, 128.72, 129.28, 131.41, 133.99, 140.70, 149.10, 150.38, 150.98, 154.09 (C=O), 154.75, 161.96 (C=O); MS: *m*/*z* (%) = M + 2, 411 (0.78), M^+^, 409 (1.5), 347 (2), 303 (3), 256 (24), 150 (79), 72 (33), 44 (100); Anal. calcd. for C_19_H_12_ClN_5_O_4_ (409.78): C, 55.69; H, 2.95; N, 17.09. Found: C, 55.84; H, 2.94; N, 17.34.

### 3.2. Biological Activity

#### 3.2.1. Anticancer Evaluation

##### Evaluation of Cytotoxic Effects of the Prepared Compounds

Mammalian cell lines: A-549 cells (human Lung cancer cell line) were obtained from VACSERA Tissue Culture Unit (Cairo, Egypt). Dimethyl sulfoxide (DMSO), crystal violet and trypan blue dye were purchased from Sigma-Aldrich Chemie GmbH (Taufkirchen, Germany). Fetal Bovine serum, DMEM, HEPES buffer solution, L-glutamine, gentamycin and 0.25% Trypsin-EDTA were purchased from Lonza (Basel, Switzerland).

Crystal violet stain (1%) was composed of 0.5% (*w*/*v*) crystal violet and 50% methanol then made up to volume with ddH_2_O.

Cell line Propagation: The cells were propagated in Dulbecco’s modified Eagle’s medium (DMEM) supplemented with 10% heat-inactivated fetal bovine serum, 1% L-glutamine, HEPES buffer and 50 µM/mL gentamycin. All cells were maintained at 37 °C in a humidified atmosphere with 5% CO_2_ and were subcultured two times a week [40,41].

Cytotoxicity evaluation using viability assay: For cytotoxicity assay, the cells were seeded in a 96-well plate at a cell concentration of 1 × 10^4^ cells per well in 100 µL of growth medium. Fresh medium containing different concentrations of the test sample was added after 24 h of seeding. Different concentrations of the tested chemical compound were added to 96-well, flat-bottomed microtiter plates (Falcon, Somerset, NJ, USA) using a multichannel pipette. The microtiter plates were incubated at 37 °C in a humidified incubator with 5% CO_2_ for of 24 h. Triplicates from each concentration were conducted. Control cells were incubated without sample. After incubation of the cells at 37 °C for 24 h, the viable cells yield was determined by a colorimetric method [42,43,44].

Briefly, media were aspirated and the crystal violet solution (1%) was added to each well for 30 min. The stain was removed, and the plates were rinsed using tap water to remove the excess stain. Glacial acetic acid (30%) was added to the wells, mixed thoroughly, and the absorbance was measured at 490 nm using a Microplate reader (SunRise, TECAN, Inc., Mannedorf, Switzerland). The results were normalized to the background absorbance as baseline in wells without stain. Treated samples were compared with the cell control in the absence of the tested compounds. The cell cytotoxic effect of each tested compound was calculated. The viability of cells were determined from the formula [(ODt/ODc)] × 100% where ODt is the mean optical density of wells treated with the tested sample and ODc is the mean optical density of untreated cells. The relation between surviving cells and drug concentration was plotted to get the survival curve of each tumor cell line for each compound. The 50% inhibitory concentration (IC_50_) is the concentration of tested compound to stop the growth of 50% of initial cells. The IC_50_ values were determined from the dose response curve of each compound with the Graph pad Prism software package (San Diego, CA, USA).

#### 3.2.2. Molecular Docking Study

The structures of all tested compounds including the co-crystalized ligands were modeled using the Chemsketch software (Toronto, ON, Canada) (http://www.acdlabs.com/resources/freeware/). The structures were optimized and energy minimized using VEGAZZ software (Milano, Italy), and saved as PDB format. Using AutoDockTools 1.5.6 (La Jolla, CA, USA), all compounds were converted to PDBQT format [45,46]. The optimized compounds were used to perform molecular docking against five proteins that represent vital targets for chemotherapeutic drugs, including cyclin dependent kinase-2 (CDK2), B-cell lymphoma 2 (BCL2), Janus kinase 2 (Jak2), p53 binding site in MDM2 (P53) and Dihydrofolate reductase (DHFR). The three-dimensional structure of the molecular target was obtained from Protein Data Bank (PDB) from the website (www.rcsb.org): CDK2 (PDB:1DI8, https://www.rcsb.org/structure/1DI8), BCL2 (PDB: 2O2F, https://www.rcsb.org/structure/2O2F), Jak2 (PDB: 5AEP, https://www.rcsb.org/structure/5AEP), P53 (PDB: 2LZG, https://www.rcsb.org/structure/2LZG), and DHFR (PDB: 4DFR, https://www.rcsb.org/structure/4DFR). The steps for receptor preparation included the removal of heteroatoms (water and ions), the addition of polar hydrogen, and the assignment of charge. The active sites were defined using grid boxes of appropriate sizes around the bound cocrystal ligands. The docking study was performed using Autodock vina (La Jolla, CA, USA) [47] and Chimera (San Francisco, CA, USA) for visualization [48]. All docking procedures and scoring were recorded according to established protocols [46,49,50,51].

## 4. Conclusions

In summary, we have developed a new, simple and convenient route for a one pot procedure or even a two or three component reaction. The reaction of 6-amino-1-benzyluracil with benzylidene acetoacetate affords pyridopyrimidine. On the other hand, the reaction of 5,6-diamino-1-2-chlorobenzyl)uracil with different aromatic aldehydes in DMF gave xanthines while the reaction of 5,6-diamino-1-(2-chlorobenzyl)uracil with different aromatic aldehydes and triethyl orthoformate under reflux condition afforded new lumazine derivatives in good yields. The newly synthesized compounds were evaluated for in vivo lung carcinoma inhibitory activity against cell line A549. Compounds **3b**, **6c**, **6d**, **6e**, **7c** and **7d** exhibited the most lung carcinoma inhibitory effect compared with the reference drug methotrexate. Molecular-docking analyses revealed that compounds **3b**, **6c**, **6d**, **6e**, **7c** and **7d** were the best docked ligands against most of the targeted proteins especially CDK2, Jak2, and DHFR proteins, as they displayed the lowest binding energies, critical hydrogen bonds and hydrophobic interactions compared to co-crystalized ligands and methotrexate.

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
