# Peer review of "Synthesis, In Silico Prediction and In Vitro Evaluation of Antitumor Activities of Novel Pyrido[2,3-d]pyrimidine, Xanthine and Lumazine Derivatives"

_molecules, 2020, doi:10.3390/molecules25215205_

Round 1

Reviewer 1 Report

Authors properly answered to all my comments.

Author Response

Dear referee;

After considering your comments and corrected all your suggestions point by point. Our great appreciation for your revision and your effort for giving these valuable remarks that have greatly improved the manuscript. we are looking forward to hearing positively about this manuscript

best regards

Dr. Samar Elkalyoubi

Reviewer 2 Report

The authors still uses one cell line. With just one cell line in which the compound worked and did not work in another cell line. SO, I still feel it needs to be rejected.

Author Response

Dear reviewer;

Thank you for your comments and we tried to answer your suggestion as follows:

  • The objective of this study was basically to explore the antiproliferative activity of Novel Pyrido[2,3‐d] Pyrimidine, Xanthine, and Lumazine derivatives, which is profoundly closely related to the structure of commercial anticancer drug “methotrexate”. Practically, methotrexate is commonly used as a commercial anticancer drug,

1) Especially against human lung cancer cell lines (A-549) (ref. 11).

2) The incidence of lung cancers in Egypt is the highest type due to various environmental conditions, sociological improper habits especially smoking and tobacco addiction.  

Thus, due to the similarity of the developed compounds with the reference drug methotrexate, this study has been focused on evaluating the anticancer activity and in silico analysis towards the human lung cancer cells.

3) The anticancer activity of the developed compounds has been already tested against MCF7 breast cancer. And from the result, the developed compound (as shown in the text page 7 lines 135- 138) has potentially promising cytotoxic activity against lung cancer more than breast cancer. Thus, human lung cancer has been used for further experimentation.

Our great appreciation for your revision and your effort for giving these valuable remarks that have greatly improved the manuscript. we are looking forward to hearing positively about this manuscript.

Best regards

Dr. Samar El-Kalyoubi

Reviewer 3 Report

I am satisfied with the correction of the manuscript and the authors' responses to my comments. Accept the article for publication as it is

Author Response

Dear referee;

After considering the comments and corrected all your suggestions point by point. Our great appreciation for your revision and your effort for giving these valuable remarks that have greatly improved the manuscript. we are looking forward to hearing positively about this manuscript.

Best regards

Dr. Samar El-Kalyoubi